# Controlled Synthesis of Au_25_ Superatom Using a Dendrimer Template

**DOI:** 10.3390/molecules27113398

**Published:** 2022-05-25

**Authors:** Hisanori Muramatsu, Tetsuya Kambe, Takamasa Tsukamoto, Takane Imaoka, Kimihisa Yamamoto

**Affiliations:** 1Laboratory of Chemistry and Life Science (CLS), Tokyo Institute of Technology, Yokohama 226-8503, Japan; muramatsu.h.ae@m.titech.ac.jp (H.M.); kambe.t.aa@m.titech.ac.jp (T.K.); tsukamoto.t.ad@m.titech.ac.jp (T.T.); timaoka@res.titech.ac.jp (T.I.); 2JST-ERATO, Yamamoto Atom Hybrid Project, Tokyo Institute of Technology, Yokohama 226-8503, Japan; 3JST-PRESTO, Kawaguchi 332-0012, Japan

**Keywords:** superatoms, dendrimers, gold clusters

## Abstract

Superatoms are promising materials for their potential in elemental substitution and as new building blocks. Thus far, various synthesis methods of thiol-protected Au clusters including an Au_25_ superatom have been investigated. However, previously reported methods were mainly depending on the thermodynamic stability of the aimed clusters. In this report, a synthesis method for thiol-protected Au clusters using a dendrimers template is proposed. In this method, the number of Au atoms was controlled by the stepwise complexation feature of a phenylazomethine dendrimer. Therefore, synthesis speed was increased compared with the case without the dendrimer template. Hybridization for the Au_25_ superatoms was also achieved using the complexation control of metals.

## 1. Introduction

Some clusters consisting of a few to several dozen metal atoms can provide discrete atom-like electron states according to DFT calculations with the Jellium model [1,2]. Metal clusters with this property, called “superatoms,” are attracting attention. Among them, a halogen superatom of Al_13_^−^ [3] and an alkali-metal superatom of Cu_8_^−^/Ag_8_^−^ [4] are well-known examples. In the latter case, electronegativity lower than the alkali metal elements (Li, Na, K, Rb, or Cs) is proposed, demonstrating a super-alkali superatom. Therefore, the development of synthesis methods for such superatoms has been a significant target of recent research.

Among these superatoms, Au clusters have been focused on as useful candidates for superatoms that can be synthesized not only in the gas phase but also in the solution phase [5,6]. Since the synthesis of thiol-protected Au clusters was reported by Brust and Schiffrin et al. [5], lots of research [6,7,8,9,10,11,12,13,14,15,16,17,18,19,20,21,22,23,24,25,26,27,28,29] has investigated them. Especially, Au_25_SR_18_ [30,31], Au_38_SR_24_ [6,32,33] and Au_144_SR_60_ [6,34,35] (R: hydrocarbons) are studied as superatomic clusters. They were purified and characterized by single crystal analysis [30,31]. In addition, the introduction of different elements, such as Pd, Pt, Cu, Ag, Cd, and Hg [23,24,27,28,36,37,38,39] into Au clusters has been reported. The mixing has been carried out by metal-exchanging. However, the synthesis of these materials often requires low-temperature conditions [30,40] or an etching reaction for several days [41] to increase the synthetic yields.

Dendrimer templates are attractive for the solution phase synthesis of metal clusters or particles since their sizes can be controlled in the state of the precursors [42,43,44,45,46,47,48,49,50,51,52,53,54,55,56,57,58]. One of the most famous dendrimers is the polyamidoamine (PAMAM) dendrimer, which is limited in accuracy for metal assembly due to its random complexation fashion. In contrast, phenylazomethine dendrimers enable atomicity control for cluster synthesis [59,60,61,62,63,64,65,66,67,68,69]. Some clusters prepared by this method are found to exhibit high catalytic activities depending on the number of atoms. In addition, the specific high activity of a Pt_17_ catalyst is also reported [70]. Superatom synthesis using this phenylazomethine dendrimer has also been reported. The synthesis and properties of Al_13_^−^ [71] and Ga_13_^−^ [72] in the dendrimer were also demonstrated. In this study, we investigated dendrimer-templating synthesis for the thiol-protected Au_25_ superatom. As the template, a phenylazomethine dendrimer with a tetraphenyl methane core part (TPM G4) was used. Due to the metal-complexation feature of the TPM G4, one atom blending was also demonstrated.

## 2. Result and Discussion

### 2.1. Controlled Assembly of AuCl_3_ on TPM G4

The TPM G4 has tetraphenylmethane in the core and a fourth-generation imine skeleton in the branches, with higher electron density in the inner layers (Appendix A). In order to use the TPM G4 as a precise template, metal salts and the imine parts of the TPM G4 are required to be connected by a 1:1 ratio. Based on the previous reports [65,66,67,68,69], the mixed solution of chloroform:acetonitrile = 1:1 was used as a solvent for the metal assembly. The stepwise coordination fashion between the dendrimer and AuCl_3_ was confirmed by UV-Vis titration (Appendix A). Isosbestic points, according to the number of imines at each layer of the TPM G4, were observed during the addition of the acetonitrile solution of AuCl_3_. The result indicates its 1:1 complexation. The observed isosbestic points at 336.5 nm (0~4 equiv.), 344.0 nm (4~12 equiv.), 332.0 nm (12~28 equiv.) and 328.0 nm (28~60 equiv.) indicate the stepwise coordination of the inner layer imines as shown in the Appendix A. These results indicate that the TPM G4 can be used as an accurate template for Au clusters.

The Au_25_PET_18_ superatom (PET: 2-phenylethanethiol: SCH_2_CH_2_Ph) was synthesized by adding thiol-ligand solution at the same time as the reduction of AuCl_3_ in the TPM G4 (Figure 1). Though this method is similar to our previously reported methods [62,67,70,72], it is different due to the addition of the thiol ligands to protect the surface of Au clusters. In addition, the solvent of NaBH_4_ was changed from methanol to ice-cold water. When NaBH_4_/methanol solution was used, Au_25_PET_18_ was not obtained. After the reactions, the solution was stirred for 3 h. Then the solvent was removed using an evaporator after filtration. The obtained dark-brown solid was purified by washing with a water/methanol mixture, and the Au clusters were extracted with acetonitrile. The obtained solution was subjected to MALDI-TOF-MS measurement in linear mode (Figure 2a). As a result, the peak corresponding to Au_25_PET_18_ (*m/z* = 7390.73) was detected with the fragment peak of [Au_21_PET_14_]^+^ (*m/z* = 6057.25) [73]. The calculated exact mass [74] of Au_25_PET_18_ (*m/z* = 7391.93) confirms the formation of the aimed superatom. UV-vis spectra of the synthesized Au_25_PET_18_ (Figure 2b) showed broad absorption from 800 nm to 600 nm and 450 nm indicating the formation of Au_25_PET_18_ [40,75,76,77]. The expected peak around 400 nm was unclear. This may be due to the overlap of the dendrimer absorption peaks with the peak top around 320 nm.

Size distribution and elemental analysis of the synthesized clusters were performed by STEM/EDS measurements (Figure 2c and Appendix A). The estimated size was 1.2 ± 0.3 nm, close to the longest interatomic distance of 11.99 Å observed by single-crystal analysis reported in the previous paper [31]. EDS mapping also indicates that Au and S are present in the same particle. XPS measurement was conducted with tetraoctylammonium bromide (TOAB) which was added to the solution before casting on highly oriented pyrolytic graphene (HOPG) (Appendix A). The observed peaks of Au 4f, which show higher binding energy than that of bulk Au, suggested the successful formation of the Au_25_ superatom [29].

### 2.2. Comparison with Existing Methods

In the previous cases, thiol protected Au clusters were generally synthesized by reducing multimeric Au-thiol complexes [30,40,41]. In these methods, only the most stable species were obtained by the etching process after forming large Au clusters. In contrast, the Au cluster was synthesized directly through the collecting process of AuCl_3_ to the dendrimer in this method (Figure 1). Therefore, the cluster formation speed is different from the previous cases [41]. MALDI-TOF-MS spectra confirmed the direct synthesis process (Figure 3a and Appendix A). The intense peak corresponding to Au_25_PET_18_ (*m/z* = 7390.73) was generated within 15 min without peaks of large Au clusters. This result suggests that the assembly process on the TPM G4 is directly responsible for the Au_25_ nucleation. After one day of reaction, the main peak was still detectable; however, the peaks of degraded clusters also appeared. Therefore, it was appropriate to control the reaction time within a few hours of synthesis using the TPM G4.

The same synthetic procedure was carried out without the dendrimer (Appendix A). This method also shows the formation of Au_25_PET_18_ after a reaction time of about 30 min (Figure 3b and Appendix A). This reaction time is considered to be derived from the etching process of the nanoparticles to the Au_25_ superatom by the ligands.

These differences in the formation process can be seen in the appearance of the solutions (Appendix A). When the dendrimer (TPM G4) was used, the color of the solution changed from red (0 min) [41] (the early stage of Au_25_PET_18_ synthesis) to brown (after 2 min) (color of Au_25_PET_18_). The color was maintained for 12 h. On the other hand, in the case without the TPM G4, the solution turned black at 0 min immediately after the reduction and remained grayish at 15 min. After 11 h, the formation of the Au_25_PET_18_ was suggested by the color of the solution; however, the solution contained a black precipitate, indicating the formation of large particles. In the case of the dendrimer (Figure 3a), several peaks with small intensity are also observed in the range of *m/z* = 3000–7500 at 15 min after reduction. These peaks are considered to be the small Au-thiol multimer and decomposed products of the TPM G4 (Appendix A).

### 2.3. Synthesis of MAu_24_PET_18_ Using the Dendrimer TPM G4 as a Template

The advantage of the TPM G4 includes metal blending ability. Here, we investigated the blending of metals using the TPM G4. In the case of previous methods, the introduction of other metals using the ligand effect was reported [20,24,27,28,36,37,38,39,78]. The blending of metals was conducted using the complexation process (Figure 4a). In this synthesis, one equivalent of Pd(CH_3_CN)_4_(BF_4_)_2_ and 24 equivalents of AuCl_3_ were coordinated to the dendrimer, and the thiol was added simultaneously with the reduction (Figure 4a). The formation of PdAu_24_PET_18_ was confirmed in the MALDI-TOF-MS spectrum (Figure 4b). The structure of the PdAu_24_PET_18_ was previously proposed as a Pd encapsulated structure (Appendix A). The MALDI-TOF-MS spectra show the formation of not only PdAu_24_PET_18_ but also Au_25_PET_18_. In fact, Au_25_PET_18_ was observed as a strong peak, and the intensity of PdAu_24_PET_18_ was about 1/13 of that of Au_25_PET_18_ This result suggests the partial reconstruction of the stable Au_25_PET_18_ structure. STEM/EDS analysis of the prepared particles confirmed the presence of Pd and Au on the same particle (Appendix A).

In the case of Pt introduction, the presence of Pt and Au on the same particle was confirmed by STEM/EDS analysis. However, the MALDI-TOF-MS spectra of the sample are not useful, unfortunately (Appendix A). The observed peak at *m/z* = 7392 is close to that of Au_25_PET_18_; therefore, we could not determine the formation of PtAu_24_PET_18_ in the MS spectra. In this case, the comparison of the MS spectra is difficult since the masses of PtAu_24_PET_18_ (*m/z* = 7390.93) and Au_25_PET_18_ (*m/z* = 7391.93) are very close (Appendix A).

## 3. Materials and Methods

### 3.1. Chemicals

The TPM G4 (synthesized by the method [59] reported previously by our group), gold(III) chloride (AuCl_3_, Sigma-Aldrich Japan, 99%), platinum(IV) chloride (PtBr_4_, Thermo Scientific, 99.99+% (metals basis), Pt 57% min), tetrakis(acetonitrile)palladium(II) tetrafluoroborate (Pd(CH_3_CN)_4_(BF_4_)_2_, Sigma-Aldrich Japan), 2-phenylethanethiol (HSC_2_H_4_C_6_H_5_, Sigma-Aldrich Japan, 98%), sodium borohydride (NaBH_4_, Kanto Chemical, >92.0% (T)), chloroform (CHCl_3_, Fujifilm, for Organic Synthesis, 99.0+% (Capillary GC)), acetonitrile (CH_3_CN, Kanto Chemical, Organics, >99.5% (GC)), toluene (C_6_H_5_CH_3_, Kanto Chemical, Organics, >99.5% (GC)), ultrapure water (Merck Millipore, generated with Milli-Q Element A10, conductivity 18.2 MΩ cm), trans-2-[3-(4-tert-Butylphenyl)-2-methyl-2-propenylidene]malononitrile (DCTB, TCI, >98.0% (GC)) and Tetraoctylammonium bromide ([CH_3_(CH_2_)_7_]_4_N(Br), TOAB, Sigma-Aldrich Japan, 98%) were used.

### 3.2. Assembly Accumulation of AuCl_3_ on Dendrimer TPM G4

Solution preparation was performed in a glove box under a nitrogen atmosphere. First, AuCl_3_ (22.7 mg, 74.7 µmol) was dissolved in acetonitrile to prepare a 2.99 mM solution. The dendrimer TPM G4 (0.380 mg, 34.4 nmol) was dissolved in chloroform and acetonitrile was added to prepare a 3.43 µM chloroform:acetonitrile = 1:1 solution. The complexation of AuCl_3_ and the TPM G4 was monitored by a UV-Vis spectrometer.

### 3.3. Synthesis of Au_25_PET_18_ Using Dendrimers

AuCl_3_ acetonitrile solution (10.4 mM, 25.0 mL) and the TPM G4 chloroform:acetonitrile = 1:1 solution (44.9 µM, 10.0mL) were prepared in a glove box under a nitrogen atmosphere. The TPM G4 solution was stirred for 1 h in the air under light-shielding conditions after the addition of 25 equivalents of the metal salt. After 1 h, ice-cold water with NaBH_4_ (26.6 mg, 0.704 mmol, 1500 equivalents of the metal salt) and 2-phenylethanethiol (10 µL, 74.5 µmol) were added for reduction and thiol protection. The amount of PET (Au:PET = 1:6) was determined based on a previous report [44]. The reaction was monitored by MALDI-TOF-MS.

After the addition of the reducing agent, the organic layer of the reaction solution was filtered. Then, the solution was concentrated using an evaporator. The obtained samples were washed several times with water/methanol. For STEM measurements, 1 µL of the sample solution (acetonitrile) was drop-casted onto a TEM grid (grid with carbon support film Super Hi-Res Carbon SHR-C075 STEM Cu75P grid, Okenshoji Co., Ltd., Tokyo, Japan). After the casting, the TEM grids were vacuum dried overnight. For XPS measurements, TOAB was added to the sample solution at 1.2 equivalents of Au. The solution was cast onto HOPG (490HP-AB HOPG SPI-3(ZYH) Grade 5 × 5 × 1 mm, Alliance Biosystems).

In this method, Au_25_PET_18_ can be synthesized by using a TPM G4 solution with concentrations of 9.29 µM–3.04 mM.

### 3.4. Synthesis of MAu_24_PET_18_ (M = Pd, Pt) Using Dendrimers

AuCl_3_ (79.4 mg, 0.262 mmol) was dissolved in acetonitrile to prepare a solution of 10.5 mM; Pd(CH_3_CN)_4_(BF_4_)_2_ (15.4 mg, 34.7 µmol) was dissolved in acetonitrile to prepare a solution of 3.47 mM; PtBr_4_ (18.8 mg, 36.4 µmol) was dissolved in acetonitrile to prepare a solution of 3.65 mM. The dendrimer TPM G4 (10.3 mg, 930 µmol) was dissolved in chloroform, and acetonitrile was added to prepare a 46.5 µM solution (chloroform:acetonitrile = 1:1). One equivalent of the metal salt of Pd or Pt was added to the dendrimer solution under atmospheric conditions and stirred for 5 min. Then, 24 equivalents of Au metal salt were added, and the mixture was stirred for 1 h under a light shield. After 1 h, ice-cold water containing NaBH_4_ (54.9 mg, 1.45 mmol) and 2-phenylethanethiol (90 µL, 671 µmol) were added simultaneously to the dendrimer complex solution. The amount of 2-phenylethanethiol was determined based on a previous report [44]. The reaction was monitored by MALDI measurements.

### 3.5. Characterization

The UV-vis absorption spectra were obtained by Shimadzu UV-3150 and UV-3600 spectrometers. MALDI-TOF-MS spectra were obtained by Bruker microflex-YI, ultrafleXtreme. DCTB was used as the matrix, and toluene or chloroform was used as the solvent at a concentration of 20 mg/mL. The samples were mixed in 1:20 volume ratios. STEM measurements were carried out by a JEOL JEM-ARM200F equipped with an EDS analyzer (acceleration voltage: 80 kV). XPS measurements were carried out by ULVAC-PHI ESCA 1700R. The X-ray source was monochrome Al Kα (1486.7 eV). The neutralizer’s emission current was 1 µA.

## 4. Conclusions

Dendrimer template synthesis of a Au_25_ superatom was developed with thiol protection. Unlike conventional synthetic methods, the conditions for synthesis are unique in that they do not require special conditions (e.g., low temperature reaction, a reaction in two separated aqueous and organic layers, or a long reaction time in the order of several days). The reduction and thiol coordination are performed for AuCl_3_ in the TPM G4, resulting in a short-time synthesis. The addition of different elements using this method was also demonstrated by using Pd or Pt atoms.

## Figures and Tables

**Figure 1 molecules-27-03398-f001:**
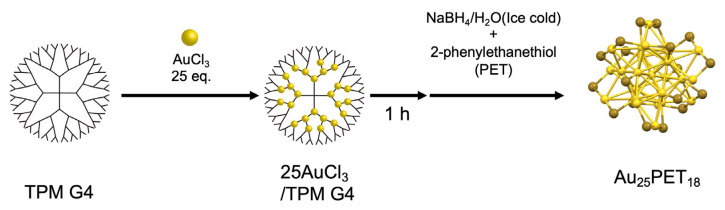
Synthetic scheme of an Au_25_PET_18_ superatom using the TPM G4 template.

**Figure 2 molecules-27-03398-f002:**
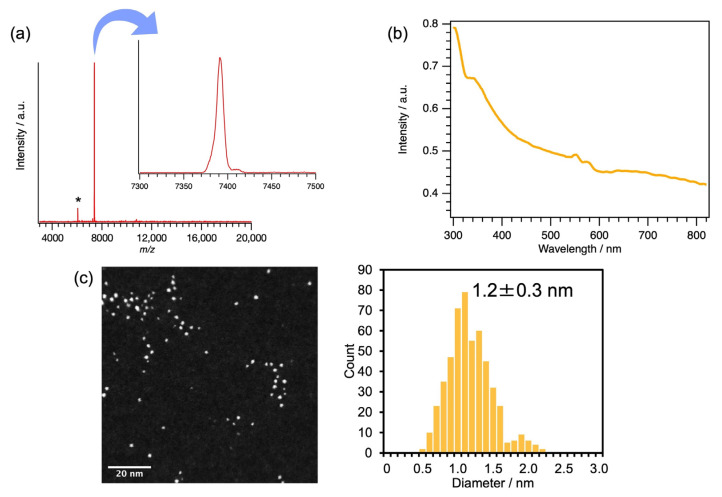
Characterization of the Au_25_PET_18_. (**a**) MALDI-TOF-MS spectra (* is a fragment peak), (**b**) UV-Vis absorption spectrum and (**c**) a STEM image and the particle size distribution of the Au_25_PET_18_ sample.

**Figure 3 molecules-27-03398-f003:**
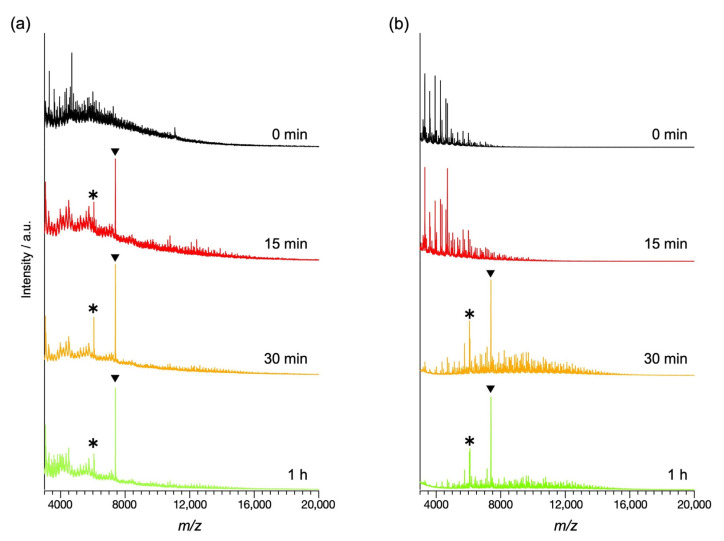
MALDI-TOF-MS spectra during Au_25_PET_18_ synthesis (**a**) with TPM G4 and (**b**) without TPM G4. ▼ are the peaks of Au_25_PET_18_, and * is the fragment peaks.

**Figure 4 molecules-27-03398-f004:**
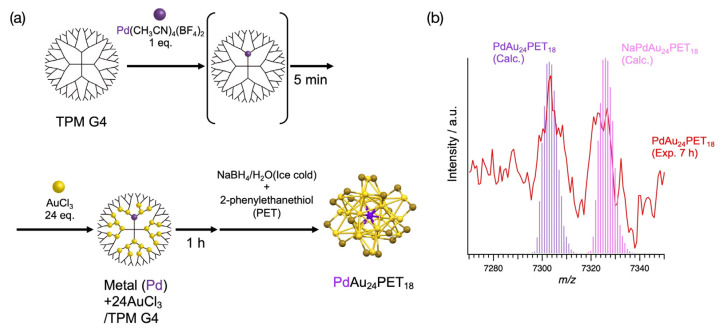
(**a**) Synthetic scheme of thiol-protected Au clusters blended with different elements (Pd) and (**b**) MALDI-TOF-MS spectrum of PdAu_24_PET_18_.

## Data Availability

The data generated and analyzed during the study are available in this article.

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
