# Peer review of "Controlled Synthesis of Au25 Superatom Using a Dendrimer Template"

_molecules, 2022, doi:10.3390/molecules27113398_

Round 1

Reviewer 1 Report

This paper describes a formation of Au25 supercluster or superatom with a dendrimer template, TPM (tetraphenylmethane) G4. The Au25 superatom was formed in the two-phase synthesis involving chloroform, methanol and water, and the synthesis was shown to be confirmed by MALDI-TOF mass and TEM; the main reducing agent was sodium borohydride. Authors also demonstrated that single palladium atom can be inserted into Au25 superatom to form PdAu24 superatom in conjunction with Au25, which is shown in MALDI spectrum.

I think this work is worthwhile to publish in this journal and recommend accepting it after minor revision. I wonder why the mass spectrum in Fig 4(b) for synthesizing PdAu24 and Au25 is so noisy compared to relatively clear one as in Fig 2(a). Does it mean that the synthetic yield of PdAu24 with Au25 can be substantially deteriorated simply by addition of 1 eq. of Pd precursor to 24 eq. AuCl3? Please elaborate this.

Minor typos.

Line 82: m/z = 73910.93?

Line 102: previously? Or in the previous case?   

Author Response

This paper describes a formation of Au25 supercluster or superatom with a dendrimer template, TPM (tetraphenylmethane) G4. The Au25 superatom was formed in the two-phase synthesis involving chloroform, methanol and water, and the synthesis was shown to be confirmed by MALDI-TOF mass and TEM; the main reducing agent was sodium borohydride. Authors also demonstrated that single palladium atom can be inserted into Au25 superatom to form PdAu24 superatom in conjunction with Au25, which is shown in MALDI spectrum.

Answer: Thank you for your comments.

I think this work is worthwhile to publish in this journal and recommend accepting it after minor revision. I wonder why the mass spectrum in Fig 4(b) for synthesizing PdAu24 and Au25 is so noisy compared to relatively clear one as in Fig 2(a). Does it mean that the synthetic yield of PdAu24 with Au25 can be substantially deteriorated simply by addition of 1 eq. of Pd precursor to 24 eq. AuCl3? Please elaborate this.

Answer: Thank you for the comments. In fact, Au25PET18 was observed as a strong peak, and the intensity of PdAu24PET18 was weaker than that of Au25PET18. Though MS intensities may not reflect synthetic yields due to the effect of ionization process, comments for the MS intensities were added to the maintext.

“In fact, Au25PET18 was observed as a strong peak, and the intensity of PdAu24PET18 was about 1/13 of that of Au25PET18.”

Minor typos.

Line 82: m/z = 73910.93?

Line 102: previously? Or in the previous case?  

Answer: Thank you very much. These points were revised.

Reviewer 2 Report

The authors presented a novel synthetic method of Au25 superstorm using a dendrimer template. The new method is faster compared with previously reported methods. And it's also facile when introducing other metal elements into the Au25 superstorm structure. The writing is easy to follow. Therefore I recommend this article to be published after minor revisions listed below:

  1. In 2.1 controlled assembly of AuCl3 on TPM G4, the calculated exact mass of Au25PET18 should be around 7391 (line 82).
  2. In line 102, "In the previously case" should be "In the previous cases".

Author Response

Review Report2

The authors presented a novel synthetic method of Au25 superstorm using a dendrimer template. The new method is faster compared with previously reported methods. And it's also facile when introducing other metal elements into the Au25 superstorm structure. The writing is easy to follow. Therefore I recommend this article to be published after minor revisions listed below:

Answer: Thank you for your comments.

  1. In 2.1 controlled assembly of AuCl3 on TPM G4, the calculated exact mass of Au25PET18 should be around 7391 (line 82).
    Answer: Thank you for your comments. We have revised the point. The MS calculated values was corrected as follows.
    “The calculated exact mass peak [72] of Au25PET18 (m/z = 7391.93) confirms the formation of the aimed superatom.”

  2. In line 102, "In the previously case" should be "In the previous cases".
    Answer: Thank you for your comments. We have revised the point.

Reviewer 3 Report

The article by K. Yamamoto et al. is devoted to the synthesis of golden-25 superatom using the dendrimer template. The cluster is thiol-protected. The method is good combination of previously known techniques. The characterization of results is proper. In addition to the synthesis of the Au25 superatom, the possibility of including other metals such as palladium and platinum in it is shown. In general, the results are ok and proper for publication in Molecules, but the main problem of this article is writing. Before publication, the text must be thoroughly checked and polished. There are a lot of unsuccessful phrases that need to be rewritten in a more scientific language, excluding the constant repetition of the word "then" (as an example lines 76-77). Many typos like the meaning of m/z = 73910.93 – line 82. In particular, you should check the captions to the pictures in supplementary materials, for example, Figure S2, where you also need to describe what are a, b, c and d. Also check the caption to Figure S13.  Figure S11 is not clear because of too much rising of signals.

Also, part of the introduction requires some refinement. The material is very unevenly distributed. There are a lot of references in the first part of it (lines 35), whereas in the second part there are clearly not enough of them (lines 49).

Also I didn’t get some moments.

Figure 3. (lines 114-116). I look at the MALDI spectra (a and b) and see that they are almost the same. The item b without TPM G4 also shows the formation of the main product along with the by-product, but only 15 minutes later. The spectra after 30 minutes and after an hour are almost identical. I see the same thing in the Figures S5 and S7. Can you clarify it?

The synthesis technique is based on the preparation of solutions of a certain concentration. It's not very convenient. Is it possible to give the synthetic method in moles, grams and milliliters and volumes of solutions?

Lines 158. It seems that the words ‘tetrafluoroboric acid’ are needless.

Why do you take exactly 7.8 eqv. of 2-phenylethanetrhiol to AuCl3? In general, could you describe the process in more detail from a chemical point of view?

Do you have any ideas why mass spectra with platinum are not good?

Author Response

P lease find attached file

Round 2

Reviewer 3 Report

The article has been substantially updated and corrected. I consider publication possible in its present form.